# Corinthian Currants Promote the Expression of Paraoxonase-1 and Enhance the Antioxidant Status in Serum and Brain of 5xFAD Mouse Model of Alzheimer’s Disease

**DOI:** 10.3390/biom14040426

**Published:** 2024-04-01

**Authors:** Dimitris Lymperopoulos, Anastasia-Georgia Dedemadi, Maria-Lydia Voulgari, Eirini Georgiou, Ioannis Dafnis, Christina Mountaki, Eirini A. Panagopoulou, Michalis Karvelas, Antonia Chiou, Vaios T. Karathanos, Angeliki Chroni

**Affiliations:** 1Institute of Biosciences and Applications, National Center for Scientific Research “Demokritos”, Agia Paraskevi, 15341 Athens, Greece; 2Department of Biology, National and Kapodistrian University of Athens, Zografou, 15784 Athens, Greece; 3Department of Chemistry, National and Kapodistrian University of Athens, Zografou, 15784 Athens, Greece; 4Laboratory of Chemistry-Biochemistry-Physical Chemistry of Foods, Department of Dietetics and Nutrition, Harokopio University, 17676 Kallithea, Greecechiou@hua.gr (A.C.); vkarath@hua.gr (V.T.K.); 5Research and Development Department, Agricultural Cooperatives’ Union of Aeghion, 25100 Aeghion, Greece; mkarvelas@pesunion.gr

**Keywords:** Alzheimer’s disease, paraoxonase-1, Corinthian currants, antioxidant effect, enzyme activity, protein expression, mouse model

## Abstract

Paraoxonase-1 (PON1), a serum antioxidant enzyme, has been implicated in Alzheimer’s disease (AD) pathogenesis that involves early oxidative damage. Corinthian currants and their components have been shown to display antioxidant and other neuroprotective effects in AD. We evaluated the effect of a Corinthian currant paste-supplemented diet (CurD), provided to 1-month-old 5xFAD mice for 1, 3, and 6 months, on PON1 activity and levels of oxidation markers in serum and the brain of mice as compared to a control diet (ConD) or glucose/fructose-matched diet (GFD). Administration of CurD for 1 month increased PON1 activity and decreased oxidized lipid levels in serum compared to ConD and GFD. Longer-term administration of CurD did not, however, affect serum PON1 activity and oxidized lipid levels. Furthermore, CurD administered for 1 and 3 months, but not for 6 months, increased PON1 activity and decreased free radical levels in the cortex of mice compared to ConD and GFD. To probe the mechanism for the increased PON1 activity in mice, we studied the effect of Corinthian currant polar phenolic extract on PON1 activity secreted by Huh-7 hepatocytes or HEK293 cells transfected with a PON1-expressing plasmid. Incubation of cells with the extract led to a dose-dependent increase of secreted PON1 activity, which was attributed to increased cellular PON1 expression. Collectively, our findings suggest that phenolics in Corinthian currants can increase the hepatic expression and activity of antioxidant enzyme PON1 and that a Corinthian currant-supplemented diet during the early stages of AD in mice reduces brain oxidative stress.

## 1. Introduction

Late-onset Alzheimer’s disease (AD) stands as the most prevalent neurodegenerative disorder, impacting millions worldwide. AD is associated with multiple etiologies and pathological mechanisms that include excessive deposition of amyloid-β (Aβ) peptide in extracellular plaques, accumulation of neurofibrillary tangles of hyperphosphorylated tau in neurons, increased neuroinflammatory responses, as well as oxidative imbalance and subsequent oxidative stress in the brain [1]. It has been proposed that oxidative stress functions as a bridge for the various pathogenetic mechanisms of AD [2,3]. Several studies have indicated that oxidative stress manifests as an early event in the disease’s progression, potentially playing a pivotal role in the neurodegeneration cascade [2,4]. Oxidative stress develops as a result of the oxidation of molecules in cells, an increase in reactive oxygen species, and a lack of sufficient antioxidant defense in the brain [2,3,4].

Paraoxonase-1 (PON1), an enzyme with antioxidant function, has been proposed to be important for oxidative balance during AD pathogenesis [5]. PON1 is a calcium-dependent enzyme produced mainly by the liver and secreted in circulation, where it is associated with high-density lipoprotein (HDL) particles [6,7]. PON1 can hydrolyze a wide variety of substrates, displaying activities of lactonase, arylesterase, and paraoxonase [5]. Hydrolysis of oxidized lipids represents the major function of this enzyme [8]. Serum PON1 activity has been consistently found to be lower in AD patients as compared to non-demented controls [5,9]. Furthermore, lower PON1 arylesterase activity has also been associated with mild cognitive impairment, suggesting that alterations in PON1 activity may be an early event in dementia progression [5]. Although the major site for PON1 expression is the liver, the detection of PON1 mRNA in the human brain [10], protein in the mouse brain [11], and enzyme activity in human cerebrospinal fluid [12] indicates a more direct role of PON1 in the brain. Of note, PON1 has been found in significant amounts in cells around Aβ plaques in several regions of the brain of the Tg2576 mouse model of AD, implicating brain PON1 in AD pathogenesis [13].

Naturally existing compounds, like plant polyphenols, have been proposed to exert potential neuroprotective benefits against AD due to several actions that include potent antioxidant effects [14]. Diets supplemented with polyphenol-rich foods have the potential to delay AD progression [15] and it is suggested that it could be most effective to start a suitable dietary intervention early in the progression of the disease [15]. In addition, polyphenol-rich foods and individual polyphenolic compounds have been shown to increase PON1 expression and activity in cell-based, animal, and human studies [16,17]. Most of these positive effects on PON1 have been observed for quercetin and resveratrol [16,17].

A variety of antioxidant polyphenolics (including quercetin and resveratrol) is present in grapes, and grape products such as raisins, which are dried grapes consumed in many places in the world [18,19]. Raisins produced from a specific variety of black grape, i.e., *Vitis vinifera* L., known as var. *Apyrena* (Corinthian currants), are predominantly cultivated in Southern Greece and those designated as Vostizza currants belong to a high-quality category, shown previously to exhibit enhanced antioxidant capacity compared to other sub-varieties [20]. In a previous study conducted in our lab, it was demonstrated that the polar phenolic extract obtained from Corinthian currants displays free-radical-scavenging properties and reduces the oxidative stress triggered by the uptake of Aβ42 by human neuroblastoma SK-N-SH cells [21]. Additionally, consumption of Corinthian currants for one month by the 5xFAD mouse model of AD reduced AD-related pathological features, such as brain Aβ42 levels and neuroinflammation, in the early stages of the disease [22].

In the current study, we examined the effect of Corinthian currant-supplemented diet on PON1 activity and levels of oxidation markers in serum and the brain of 5xFAD mice. In this mouse model of AD, intraneuronal accumulation of Aβ is observed at 1.5 months of age, along with amyloid deposition in the cortex and subiculum, and neuroinflammation that are evident by 2 months [23]. Thus, we started the diet intervention during the early stages of the disease (in 1-month-old mice) and assessed the impact of the currant-supplemented diet, administered for 1, 3, and 6 months, as compared to a normal diet or sugar-matched diet, on PON1 activity and oxidation marker levels in serum and the brain of male and female mice. In addition, this dietary protocol allowed the examination of age-dependent changes in PON1 activity and oxidation marker levels in serum and brain of mice. Furthermore, we examined the effect of currant phenolics on cellular PON1 expression and activity by treating Huh-7 hepatocytes and HEK293 cells transfected with a PON1-expressing plasmid with Corinthian currant polar phenolic extract.

## 2. Materials and Methods

### 2.1. Animals

The 5xFAD transgenic mouse model of AD, on a C57BL/6 background, carries and inherits familial AD mutations of human amyloid precursor protein (APP) [Swedish (K670N, M671L), Florida (I716V) and London (V717I)] and presenilin-1 (M146L and L286V) [23,24]. Genotyping of 5xFAD mice was conducted via PCR analysis of tail DNA [22], with the mice maintained in a heterozygous state. The animals were housed under standard conditions (24 °C, 12 h light/dark cycle, lights on at 8:00 a.m.) with access to food and water ad libitum. All procedures involving animals were conducted in accordance with European legislation and were approved by the Animal Care and Use Committee of the National Center for Scientific Research “Demokritos” and by the Directorate of Agricultural and Veterinary Policy of the Region of Attica (Animal welfare assurance number: EL25 BIO 039, 6806/21-12-2018).

### 2.2. Preparation of Currant Paste and Extract

Currant paste was supplied by the Agricultural Cooperatives’ Union of Aeghion, Greece. It was prepared as described previously [22,25], using Corinthian currants produced from a special variety of black grape, i.e., *Vitis vinifera* L., var. *Apyrena*, originating from the Aeghion (Vostizza) area in the Peloponnese in Southern Greece.

To extract polar phenolic compounds from currants, freeze-dried samples were treated with methanol, following a previously described procedure [21,26]. Aliquots of methanolic extract were stored under nitrogen at −20 °C. The total polar phenolic content of the extract was determined using the Folin Ciocalteu assay, with gallic acid serving as the reference standard. The concentration of the currant polar phenolic extract is expressed in milligrams of gallic acid equivalents (GAE) per milliliter of methanolic extract.

### 2.3. Animal Study Protocol

At four weeks of age, male and female transgenic mice were randomly allocated into nine groups, each receiving one of three types of diet for a duration of 1, 3, and 6 months. This comprised a total of 99 mice, with 6 males and 5 females per diet and time point, as described previously [22]. Specifically, the mice were fed either a standard chow diet; a chow diet enriched with 5% (*w*/*w*) currants paste; or a chow diet containing 1.75% (*w*/*w*) glucose/1.75% (*w*/*w*) fructose, adjusted to match the sugar content in currants [27]. Upon completion of each feeding period, the mice were weighed and then sacrificed by decapitation. Approximately 1 mL of blood was obtained from each animal via cardiac puncture using 26G 3/8 needles (BD Biosciences, Heidelberg, Germany) into standard 1.5 mL Eppendorf tubes. The blood was left at room temperature for 1 h to facilitate blot clotting, after which serum was collected at the supernatant following centrifugation of samples at 1500× *g* for 15 min at 4 °C. The serum was then stored in aliquots at −80 °C. Furthermore, the brain of each mouse was removed, rinsed with ice-cold phosphate-buffered saline, and hemisected. The cortical region was dissected out from the hemispheres, rapidly frozen in liquid nitrogen, and stored at −80 °C.

### 2.4. Preparation of Cortical Homogenates

Each cortical sample underwent homogenization with fifteen strokes using a glass–Teflon homogenizer in 500 μL of buffer A (50 mM Tris, 150 mM NaCl, containing 20 mM NaF, 1 mM Na_3_VO_4_, 1 mM 4-(2-aminoethyl) benzenesulfonyl fluoride hydrochloride, complete mini protease inhibitor cocktail (Roche, Mannheim, Germany) and phosphatase inhibitors PhosSTOP (Roche), pH 7.5). Subsequently, aliquots of the homogenized cortical samples were diluted 1:3 with buffer A and submitted to additional homogenization by sonication (Ultrasonic Processor UP200S, Heilscher Ultrasonics, Teltow, Germany) utilizing a 3 mm diameter probe at 50% per second pulse mode and 30% sonic power, for 20 s × 3 with 40 s intervals between sonication periods. The samples were then stored at −80 °C until further use. Protein levels in the homogenate samples were quantified using the BCA protein assay kit (Pierce, Rockford, IL, USA).

### 2.5. Measurement of apoA-I Levels in Serum

The levels of serum apoA-I were determined using the Elisa pro mouse apoA-I kit (Mabtech, Nacka Strand, Sweden) according to the manufacturer’s instructions.

### 2.6. Measurement of Paraoxonase Activity of PON1 in Serum

The paraoxonase (PON) activity of PON1 was assessed using paraoxon as the substrate [28]. In brief, the assays were conducted in a final volume of 250 μL, comprising 5 μL of serum, 5.61 mM paraoxon (diethyl p-nitrophenyl phosphate), 100 mM Tris-HCl, and 2 mM CaCl_2_, pH 8.0. The rate of p-nitrophenol formation resulting from paraoxon hydrolysis was determined by recording the increase in absorbance at 405 nm over 38 min at room temperature using a plate reader spectrophotometer (Spark, Tecan Trading AG, Zurich, Switzerland). The serum PON activity of PON1 was normalized with the apoA-I concentration and expressed as U/mg of apoA-I, where one U represents the activity catalyzing the formation of one μmol of p-nitrophenol per minute.

### 2.7. Measurement of Arylesterase Activity of PON1 in Serum and Cortical Homogenates

The arylesterase (ARE) activity of PON1 was measured using phenyl acetate as the substrate [28]. In brief, the assays were conducted in a final volume of 250 μL, comprising 5 μL of serum diluted 1:50, or 5 μL of cortical homogenate, 1 mM phenyl acetate, 100 mM Tris-HCl, and 2 mM CaCl_2_, pH 8.0. The rate of phenol formation resulting from phenyl acetate hydrolysis was determined by recording the increase in absorbance at 270 nm over 5 min at 25 °C using a microtiter plate reader (Spark, Tecan Trading AG). The serum ARE activity of PON1 was normalized with the apoA-I concentration and expressed as kU/mg of apoA-I, where one U represents the activity catalyzing the formation of one μmol of phenyl acetate per minute. The ARE activity of PON1 in cortical homogenates was normalized with the protein concentration and expressed as U/g of protein.

### 2.8. Measurement of Malondialdehyde (MDA) Levels in Serum

The levels of MDA, a common lipid peroxidation product, were measured in serum by the Thiobarbituric Acid Reactive Substances (TBARS) assay [29] with some modifications. A volume of 20 μL of serum was mixed with 500 μL of 42 mM H_2_SO_4_ aqueous solution and 125 μL of 10% *w*/*v* phosphotungstic acid aqueous solution, incubated for 5 min at room temperature, and centrifuged at 13,000× *g* for 3 min. After discharging the supernatant, the pellet was suspended on ice in 200 µL of solution containing 0.05 *w*/*v* butylated hydroxytoluene in water: methanol 99:1. Standard MDA aqueous solutions at concentrations of 0.2–4 µM were also prepared using a 500 μM MDA (Cayman Chemical, Ann Arbor, MI, USA) stock solution. For the assay reaction, 200 μL of each sample or standard was mixed with 600 μL of 0.335% *w*/*v* TBA (thiobarbituric acid, Cayman Chemical) in water: glacial acetic acid 7:3 and then incubated at 95 °C for 1 h. Subsequently, the mixtures were cooled in an ice bath for 10 min to stop the reaction. TBARS of each sample and standard were extracted by adding 300 μL of 1-butanol and 100 μL of NaCl 5 M, followed by centrifugation at 16,000× *g* for 3 min. Finally, 250 µL of the upper butanol phase, which contained the MDA equivalents (TBARS), were placed in wells of a 96-well plate and fluorescence was measured using a microtiter plate reader (Infinite M200, Tecan Trading AG) at 560 nm after excitation at 530 nm. Serum MDA levels were expressed in nmol/mL.

### 2.9. Measurement of Free Radical Levels in Cortical Homogenates

A volume of 10 µL of the cortical homogenates mixed with 185 µL of phosphate-buffered saline (pH 7.4) and 5 μL of 2′,7′-dichlorodihydrofluorescein (DCFH, Invitrogen/Thermo Fisher Scientific, Paisley, UK) 1 mM was incubated at 37 °C for 30 min. Upon exposure to oxidants, non-fluorescent DCFH undergoes oxidation to produce fluorescent DCF (2′,7′-dichlorofluorescein). The fluorescence intensity was measured in a plate reader spectrophotometer (Infinite M200, Tecan Trading AG) at 520 nm after excitation at 480 nm. The recorded DCF fluorescence values were normalized with the protein concentration of the cortical homogenate and expressed as AU (arbitrary units) per µg of protein [30].

### 2.10. Culture of Cells and Transfection Procedure

Human hepatocytoma Huh-7 cells were grown in Dulbecco’s Modified Eagle Medium (DMEM) supplemented with 10% (*v*/*v*) fetal bovine serum (FBS) and 100 U/mL penicillin and streptomycin. The cells were seeded onto 24-well plates at a density of 6 × 10^4^ cells/well and treated with the Corinthian currant polar phenolic extract at concentrations of 1 and 5 μg GAE/mL in complete medium for 48 h. The final methanol concentration in each well was 0.68% (*v*/*v*). Control cells were also included, treated without the extract but in the presence of 0.68% (*v*/*v*) methanol. Following the 48 h incubation period, the medium was replaced with fresh medium containing heat-inactivated FBS (90 min at 56 °C to deactivate serum PON1 activity) and the cells were incubated for an additional 24 h period. Subsequently, the cell medium and cells were collected separately. The Huh-7 cells were lysed at 4 °C in lysis buffer (containing 50 mM Tris–HCl (pH 7.4), 150 mM NaCl, 1 mM EDTA, 0.1% (*v*/*v*) SDS, 0.25% (*w*/*v*) deoxycholate, and 1% (*v*/*v*) Triton-X). The protein concentration in the cell lysates was determined using the BCA protein assay kit (Pierce). The secreted PON1 arylesterase activity was measured using 100 µL of cell medium and monitoring the increase in absorbance at 270 nm for 15 min, following phenyl acetate hydrolysis, as described above. The results were normalized to the protein concentration of the cells and expressed as ΔOD_270_/min/mg of protein.

Human embryonic kidney (HEK) 293 cells grown in DMEM (high glucose), 2 mM L-glutamine, 10% (*v*/*v*) FBS, and 100 U/mL penicillin–streptomycin, were seeded onto 12-well plates at a density of 1.6 × 10^5^ cells/well and transfected with the pcDNA3.1/C-(K)-DYK/PON1 vector or with empty vector (mock). The PON1_OHu26671D_pcDNA3.1-C-(K)-DYK vector containing the human cDNA for expression of PON1 fused with a C-terminal DYKDDDDK tag was purchased from GenScript, Piscataway, NJ, USA. Transfection was performed using the jetPRIME transfection reagent (Polyplus, Illkirch, France) following the manufacturer’s instructions. Four hours after the DNA/transfection reagent addition, the medium was aspirated and cells were incubated with the Corinthian currant polar phenolic extract at concentrations of 1–10 μg GAE/mL in complete medium for 44 h. Following this incubation period, the medium was withdrawn and replaced with fresh medium containing heat-inactivated FBS for an additional 24 h incubation period. Finally, the cell medium and cells were collected separately. The HEK293 cells were lysed as described for the Huh-7 cells and cell lysate protein levels were quantified using the BCA protein assay kit (Pierce). The secreted PON1 arylesterase activity was measured using 50 µL of cell medium and monitoring the increase in absorbance at 270 nm for 10 min, following phenyl acetate hydrolysis, as described above. The results were normalized to the protein concentration of the cells and expressed as ΔOD_270_/min/mg of protein.

### 2.11. In Vitro Activation of Recombinant PON1 by Currant Extract

The arylesterase activity of recombinant PON1 was measured either in the absence or presence of currant extract at various concentrations (1–10 μg GAE/mL). As the enzyme source, we used 50 µL of cell medium of HEK293 cells transfected, in a 12-well plate, with the pcDNA3.1+/C-(K)-DYK/PON1 vector and cultured in the absence of currant extract, as described above. The arylesterase activity of recombinant PON1 was determined by monitoring the increase in absorbance, following phenyl acetate hydrolysis, at 270 nm for 10 min.

### 2.12. Detection of Cellular PON1

An amount of 30 μg of total protein from the lysates of the transfected HEK293 cells was electrophoresed on 12% SDS-PAGE gels and subsequently transferred onto nitrocellulose membranes for immunoblotting. PON1 was detected using the mouse anti-DYKDDDDK Tag monoclonal antibody 9A3 (1:500, Cell Signaling Technology, Danvers, MA, USA) and a goat anti-mouse IgG-HRP (1:1000, Cell Signaling Technology). In addition, α-tubulin was detected by a rabbit anti-α-tubulin polyclonal antibody (1:1000, Cell Signaling Technology) and a goat anti-rabbit IgG-HRP (1:2000, Merck, Darmstadt, Germany). Quantification of protein levels was conducted by densitometry analysis of the immunoreactive bands using the ImageJ software (version 1.54d) [31].

### 2.13. Cell Viability Assay

The cell viability of the Huh-7 and HEK293 cells in the presence of Corinthian currant polar phenolic extract was measured by the 3-(4,5-dimethylthiazol-2-yl)-2,5-diphenyltetrazolium bromide (MTT) assay, as described previously [21]. In brief, the cells were seeded on 96-well plates at a density of 2 × 10^4^ cells/well in complete medium. The following day, the medium was removed and the cells were treated in the absence or presence of Corinthian currant polar phenolic extract at concentrations ranging from 1 to 10 μg GAE/mL in complete medium for 48 h. After this incubation period, the medium was aspirated and the cells were incubated in serum-free medium containing 0.65 mg/mL MTT for 3 h at 37 °C. Subsequently, the medium was discarded and the dark blue formazan crystals produced by the cells were dissolved in DMSO. The absorbance was then measured at 550 nm using a microplate spectrophotometer (Infinite M200, Tecan Trading AG).

### 2.14. Statistical Analysis

Data are expressed as mean ± SD. Statistical comparisons between two groups were analyzed for significance by unpaired two-tailed Student’s *t*-test using the GraphPad Prism 8.0 software. Correlations between variables were examined using Pearson’s correlation coefficient for parametric data following the confirmation of the normality of distribution for each studied parameter by the Shapiro–Wilk test. A *p* value < 0.05 was considered statistically significant.

## 3. Results

### 3.1. Effect of Age, Sex, and Currant Diet on PON1 Activity and Lipid Peroxidation Product Levels in Serum of 5xFAD Mice

One-month-old 5xFAD mice (6 males and 5 females per group) were fed a typical diet [control (Con) group], a diet supplemented with 5% (*w*/*w*) Corinthian currant paste [currant (Cur) group], or a diet containing 1.75% (*w*/*w*) glucose/1.75% (*w*/*w*) fructose [(G/F) group], that matches the sugar content in currants, for a duration of 1, 3, and 6 months. A previous study from our lab showed that currant consumption by 5xFAD mice for 3 months resulted in lower brain Aβ42 levels in male mice and reduced neuroinflammation in both male and female mice, but further currant consumption for up to 6 months had no beneficial effect [22]. The mice did not exhibit an increased appetite in response to a sweet diet, or hyperglycemia or hypercholesterolemia at any age [22]. In the current study, we extended our studies on the effect of currant consumption in AD-related pathogenic processes by examining whether a currant-supplemented diet, as compared to control and sugar-matched diets, could affect the activity of the antioxidant enzyme PON1 and levels of oxidation markers in serum and the brain of the same groups of 5xFAD mice studied previously [22]. Additionally, we evaluated whether there were age-dependent changes in PON1 activity and oxidation marker levels in serum and the brain of the 5xFAD mice.

Both arylesterase (ARE) and paraoxonase (PON) activities of PON1 were measured in the 5xFAD mice serum using phenyl acetate and paraoxon as substrates, respectively. Paraoxon is considered the most specific substrate for measuring PON1 activity in serum, but phenyl acetate is one of the best substrates for this enzyme [7] and most of the studies examining the association of serum PON1 activity with AD have measured the arylesterase activity of the enzyme [9]. Since PON1 is associated in circulation with apoA-I-containing HDL particles [6], we measured serum apoA-I levels (Appendix A) and normalized the PON1 activity values with the apoA-I concentration. Among the mice, older mice had higher levels of apoA-I than younger mice (Appendix A) and male mice had higher apoA-I levels than female mice (Appendix A), as reported before [32]. There was, though, no diet-dependent effect, as no significant difference in serum apoA-I levels was observed among the three groups at any time point.

Previous studies have shown that AD progression has been associated with a lower serum PON1 activity [5]. Therefore, we evaluated the effect of aging, accompanied by disease progression, on serum PON1 activity of the 5xFAD mice fed with the control diet. Our analysis showed an age-dependent reduction of PON1 activity for both male and female mice. Specifically, at 3 and 6 months both arylesterase and paraoxonase activities were reduced as compared to the respective activities at 1 month (Figure 1A,B and Appendix A). Comparison of arylesterase and paraoxonase activities between male and female 5xFAD mice in all age groups showed that both activities were higher in female mice (Figure 1C and Appendix A), in accordance with the same observation reported previously for serum PON1 activity in several mouse strains [33].

Since the major function of PON1 is the hydrolysis of oxidized lipids [8], we proceeded to determine the serum malondialdehyde (MDA) levels, as a measure of lipid peroxidation levels [34]. We observed an age-dependent increase of MDA levels for both male and female mice, with the lowest values to be detected at 1 month (Figure 2A,B). Furthermore, male 5xFAD mice had higher serum MDA levels than female mice at any age (Figure 2C). Of note, serum MDA levels showed an inverse correlation with arylesterase and paraoxonase activities of PON1 in mice of both sexes (Appendix A).

An evaluation of the effect of the currant-supplemented diet on serum PON1 activity, showed that after 1 month the currant group mice of both sexes had higher arylesterase (Figure 3A,D) and paraoxonase activity (Appendix A) as compared to the control and G/F groups. Longer currant supplementation, for 3 and 6 months, did not, however, show a beneficial effect on the serum PON1 activity of mice (Figure 3B,C,E,F and Appendix A). Similarly, 5xFAD mice of both sexes that received the currant diet for 1 month, but not for 3 or 6 months, had lower serum MDA levels, as compared to mice that received the control or G/F diet (Figure 4).

Taken together, these findings demonstrate that a currant-supplemented diet displays antioxidant effects in the serum of young 5xFAD mice when administered for a short period (1 month). Consequently, we proceeded to examine whether the currant diet could similarly impact the antioxidant status in the brain of these mice.

### 3.2. Effect of Age, Sex, and Currant Diet on PON1 Activity and Free Radical Levels in the Brain of 5xFAD Mice

Measurement of the arylesterase activity in the cortical homogenates of mice fed the control diet demonstrated an age-dependent reduction in the following order: 1 month > 3 months > 6 months for male 5xFAD mice and 1 month ≃ 3 months > 6 months for female mice (Figure 5A,B). Furthermore, female 5xFAD mice at 3 and 6 months displayed higher arylesterase activity in their cortex as compared to male mice of the same age (Figure 5C). Determination of free radical levels in cortical homogenates, assessed by measurement of DCF fluorescence intensity, showed an age-dependent increase in the following order: 1 month < 3 months < 6 months for male 5xFAD mice and 1 month ≈ 3 months < 6 months for female mice (Figure 6A,B). Comparison of free radical levels between male and female mice showed that the levels were lower in female mice at 3 and 6 months (Figure 6C). The free radical levels were found to correlate inversely with PON1 arylesterase activity in the cortical homogenate of mice of both sexes (Appendix A).

Evaluation of the effect of the currant-supplemented diet on the antioxidant status in the brain showed that administration of the currant-supplemented diet for 1 and 3 months resulted in increased arylesterase activity and reduced free radical levels in the cortex of 5xFAD mice of both sexes (Figure 7A,B,D,E and Figure 8A,B,D,E). Further administration, though, of currant-supplemented diet, up to 6 months, had no effect on arylesterase activity and free radical levels in the cortex of 5xFAD mice (Figure 7C,F and Figure 8C,F). In summary, our analyses indicate that a diet supplemented with currants for a duration of up to 3 months may alleviate the oxidative burden in the brains of 5xFAD mice.

### 3.3. Effect of Currant Extract on the Activity of PON1 Secreted from Cells

Several protective effects of currants against pathogenic processes related to diseases have been attributed to their phenolic components [20,21,22,26,35]. Furthermore, various polyphenolic compounds have been shown to increase PON1 expression and activity in cell-based studies [17]. Thus, to investigate the mechanism for the observed increase in PON1 activity in mice fed with the currant-supplemented diet we examined the effect of Corinthian currant polar phenolic extract on PON1 activity secreted by hepatocytes, the major source of the enzyme in organisms. We incubated the human hepatocytoma Huh-7 cells in the absence or presence of currant polar phenolic extract at concentrations (expressed as μg of GAE per mL) shown previously to prevent the increase of cellular oxidative stress and inflammatory responses [21,22]. At first, we examined the effect of currant extract on cell viability. Our analysis showed that there was no statistically significant effect in the viability of Huh-7 cells treated with 1 and 5 µg GAE/mL of extract (Figure 9A). A higher concentration of extract (10 μg GAE/mL), though, decreased the cell viability by ~15% (Figure 9A). The decline in cell viability as the concentrations of currant phenolic extract increased aligns with earlier findings indicating that phenolic compounds typically display bimodal pharmacological effects, demonstrating therapeutic properties at lower concentrations but inducing toxicity at higher concentrations [36]. The arylesterase activity of PON1 that was secreted in the cell medium of Huh-7 cells incubated with the currant extract at 5 and 10 µg GAE/mL was found to significantly increase as compared to the enzyme activity from untreated control cells (Figure 9B). Specifically, currant extract at a concentration of 5 µg GAE/mL induced an increase of cell-secreted arylesterase activity by ~88% compared to control cells, whereas the extract at a concentration of 10 µg GAE/mL also induced an increase in arylesterase activity, but a smaller one. The smaller increase in arylesterase activity that was induced by 10 µg GAE/mL extract could be attributed to the reduced viability of cells in the presence of such a concentration of extract.

Despite the apparent increase in PON1 arylesterase activity in the cell medium of Huh-7 hepatocytes, the absolute activity values were modest, posing challenges in the measurement of the dose-dependent effect of currant polyphenolic extract on PON1 arylesterase activity. Therefore, we proceeded to use HEK293 cells transfected with a human PON1 expression plasmid that enabled us to measure elevated PON1 activity values in the cell medium. Evaluation of the effect of currant extract on HEK293 viability showed that, similarly to the Huh-7 cells, the extract at concentrations of 1 and 5 µg GAE/mL did not affect the cell viability, but at a concentration of 10 μg GAE/mL decreased the cell viability by ~25% (Figure 10A). Measurement of arylesterase activity in the medium of HEK293 cells expressing the human PON1, in the presence of increasing concentrations of currant extract, showed that the enzyme activity was increased by all concentrations of extract that were used (Figure 10B). Specifically, the highest increase (~6-fold) was observed in the presence of 5 μg GAE/mL extract. A smaller increase in arylesterase activity was induced by 10 µg GAE/mL extract, as compared to 5 µg GAE/mL, possibly due to the reduced viability of cells incubated with this concentration of extract. As expected, the arylesterase activity in the medium of HEK293 cells transfected with an empty vector (mock) was minimal and no increase was observed in the presence of currant extract (Figure 10B).

In subsequent analyses, we examined how the currant extract increased PON1 activity in the cell medium. Firstly, we evaluated whether the currant polar phenolic extract, through its components, could directly induce the activation of PON1. For this purpose, we incubated the recombinant human PON1 with increasing concentrations of currant extract in vitro. As an enzyme source, we used the cell medium of HEK293 cells that were transfected with the PON1-expressing vector and incubated in the absence of currant extract. As shown in Figure 10C, there was only a slight increase (up to ~10%) of PON1 arylesterase activity in the presence of currant extract. This small increase, though, could not explain the observed large increase, up to ~6-fold, in the activity of secreted PON1 from cells treated with the currant extract (Figure 10B).

Subsequently, we evaluated the effect of currant polar phenolic extract on cellular PON1 expression. Analysis by immunoblotting showed that the recombinant human PON1 expressed in HEK293 cells following transfection with a PON1-expressing plasmid displayed one band of ~43 kDa and another band of ~39 kDa (Figure 10D). This pattern is similar to the pattern of PON1 detected in human plasma and the higher-molecular-mass band corresponds to glycosylated PON1 [37]. Densitometric analysis of the immunoblots showed that the currant extract, at all the concentrations used, promoted a significant increase (3.2–8.8-fold) in cellular PON1 protein levels when compared with the untreated cells (Figure 10D,E). Of note, the increase in cellular PON1 protein levels paralleled the increase in PON1 arylesterase activity in the cell medium (Figure 10B,E), suggesting that the currant extract enhances the protein expression of PON1 and subsequently secretion of the enzyme, resulting in increased activity in the cell medium.

## 4. Discussion

The current findings illustrate that the administration of a diet supplemented with Corinthian currants in 5xFAD mice of both sexes increased serum PON1 activity during the early stages of AD. Interestingly, it was found that PON1 activity also increased in the brain of mice in the early stages of the disease (up to 4 months of age). PON1 is highly expressed in the liver and then secreted to circulation [7], but previous studies showed the presence of PON1 protein in the mouse brain [11], as well as of PON1 mRNA in the human brain [10] and enzyme activity in human cerebrospinal fluid [12]. Whether the presence of PON1 in the brain is due to PON1 being transported there from the circulation by crossing the blood–brain barrier or to local synthesis is currently unclear. However, the presence of PON1 in the brain, shown previously [10,11,12] and in the current study, indicates that PON1 has a direct role in this tissue, possibly by displaying antioxidant effects.

Our analyses showed an inverse correlation of brain PON1 activity with free radical levels. Furthermore, the increase in PON1 activity following administration of the currant diet was accompanied by a reduction in free radical levels in the brain of 5xFAD mice. These findings suggest that PON1 has an important antioxidant effect on the brain during AD, which can be enhanced by dietary interventions. The brain, as a tissue with a high lipid content and high oxygen consumption rate, has been shown to be vulnerable to oxidation of its lipids and oxidative injury [2]. Levels of reactive oxygen species increase with aging and have been linked to mechanisms underlying cognitive aging and neurodegenerative diseases such as AD [2,38]. Oxidative stress has been implicated as an early and prominent feature of AD pathogenesis by post-mortem in vitro and preclinical in vivo studies [39]. Furthermore, increased lipid peroxidation has been reported to occur before the amyloid plaque formation in the 3xTg-AD mouse model of AD [40]. In addition to neuronal oxidative stress, vascular oxidative stress has been also suggested to be related to AD pathogenetic processes [41,42]. Whether vascular oxidative stress precedes neuronal oxidative stress in AD or is a consequence of AD pathogenesis has not been elucidated. Our findings showed a reduction in serum PON1 activity in 5xFAD mice aged older than 2 months, an age at which these mice start to display amyloid deposition in the cortex and subiculum, and increased neuroinflammation [23]. These findings could indicate an association of serum PON1 activity with AD pathogenesis and are in accordance with human studies showing that serum PON1 activity is lower in AD patients compared to non-demented controls [5,9].

Natural antioxidants, such as polyphenolic compounds found in plant-based foods, are being evaluated for their protective effects against AD pathogenesis through antioxidant functions [4]. Raisins, naturally sun-dried grapes, are a rich source of phenolic compounds, displaying beneficial effects for human health [19]. Raisin varieties include dark raisins, golden raisins, sultanas, and Corinthian currants. Due to their low-to-moderate glycemic index, raisins are considered a nutritious option for a snack [19]. Corinthian currants are rich in polyphenols [20,21], and their polar phenolic extract has been shown previously to exert antioxidant properties in vitro [20], as well as to reduce the oxidative stress induced by Aβ42 uptake by human neuroblastoma SK-N-SH cells [21]. Furthermore, consumption of Corinthian currants for one month by young 5xFAD mice reduced brain Aβ42 levels and neuroinflammation [22]. Additionally, in a rat model of AD, currant administration for two months was also shown to display beneficial effects by protecting against oxidative stress and spatial memory impairment [43]. Specific Corinthian currant phenolic compounds, such as quercetin and resveratrol [21], have been shown to increase PON1 expression and activity in hepatocytes [17,44,45], and diets enriched in grape polyphenolic concentrate, quercetin or resveratrol increased serum PON1 activity in various mice models [17,46]. It is, therefore, possible, based on the findings of the current study, that the Corinthian currant phenolic components increase hepatic PON1 expression in 5xFAD mice, leading to an increase of PON1 activity in the circulation, and subsequently in the brain, during the early stages of AD.

Previous cell-based studies showed the ability of polar phenolics to cross the blood–brain barrier, suggesting their potential for exerting antioxidant effects in the brain and alleviating AD [47,48]. Of note, a recent study showed the accumulation of polar phenolics in the brain of rats after administration of a Corinthian currant-supplemented diet [49]. Additional studies are required to ascertain whether Corinthian currant polar phenolics can be found also in the brain of 5xFAD mice and whether they can activate PON1 locally.

Accumulating evidence has indicated a strong interrelation between oxidative stress and inflammation in the brain and suggested that antioxidants can play a protective role by mitigating both oxidative stress and neuroinflammation in AD [50,51]. Of note, the Corinthian currant diet that was found previously to attenuate neuroinflammation in 5xFAD mice of both sexes when administered for 3 months [22] was also found in the present study to reduce oxidative stress in the brain when administered for the same period. These findings support the connection of oxidative stress and neuroinflammation and suggest that a Corinthian currant-supplemented diet may exert beneficial effects against several AD-related pathogenic processes.

Overall, our findings suggest that dietary Corinthian currant administration in the AD mouse model 5xFAD reduces the oxidative stress in the circulation and the brain at the early stages of the disease (up to 4 months of age). However, during later stages of the disease, when AD pathology has significantly impacted the brain, no beneficial effects are observed. Collectively, our cell-based and animal studies demonstrate promising results regarding the effect of currants and their components against AD, encouraging further investigation of currant consumption in interventional studies involving human subjects. Considering the current findings, and given that oxidative stress is proposed to start early in the disease progression [2,4], any intervention with Corinthian currants should start in the early stages of AD pathogenesis, prior to the onset of significant neurodegeneration.

## Figures and Tables

**Figure 1 biomolecules-14-00426-f001:**
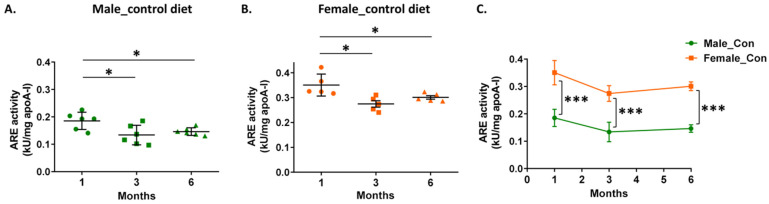
Effect of aging and sex on PON1 arylesterase (ARE) activity in serum of 5xFAD mice fed with the control diet. ARE activity was normalized with apoA-I concentration and expressed as kU/mg of apoA-I. Panel (**A**) shows values for male mice, panel (**B**) shows values for female mice, and panel (**C**) shows a comparison of ARE activity between male and female mice. Data are presented as mean ± SD (*n* = 6 males, 5 females per condition). * *p* < 0.05, *** *p* < 0.0001.

**Figure 2 biomolecules-14-00426-f002:**
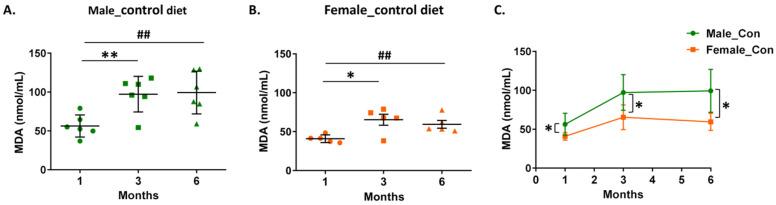
Effect of aging and sex on malondialdehyde (MDA) levels in serum of 5xFAD mice fed with the control diet. Panel (**A**) shows values for male mice, panel (**B**) shows values for female mice, and panel (**C**) shows a comparison of MDA levels between male and female mice. Data are presented as mean ± SD (*n* = 6 males, 5 females per condition). * *p* < 0.05, ^##^ *p* < 0.01, ** *p* < 0.005.

**Figure 3 biomolecules-14-00426-f003:**
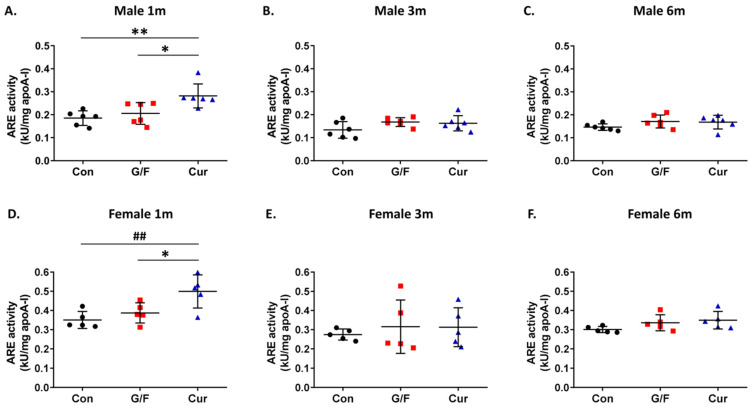
Time-dependent effect of currant diet on PON1 arylesterase (ARE) activity in serum of 5xFAD mice. ARE activity was normalized with apoA-I concentration and expressed as kU/mg of apoA-I. Panels (**A**–**C**) show values for male mice and panels (**D**–**F**) show values for female mice. Data are presented as mean ± SD (*n* = 6 males, 5 females per condition). * *p* < 0.05, ^##^ *p* < 0.01, ** *p* < 0.005.

**Figure 4 biomolecules-14-00426-f004:**
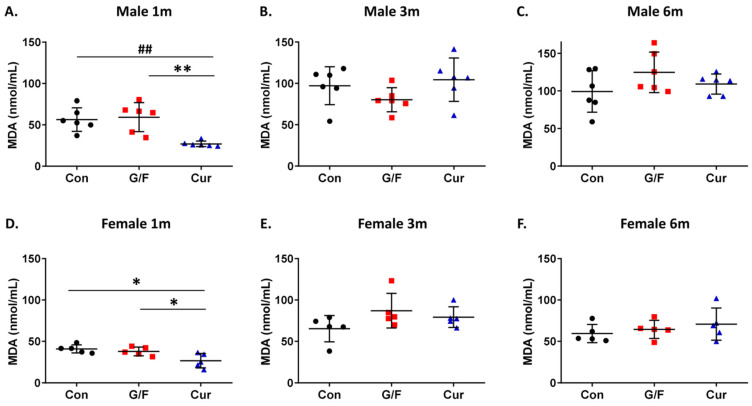
Time-dependent effect of currant diet on malondialdehyde (MDA) levels in serum of 5xFAD mice. Panels (**A**–**C**) show values for male mice and panels (**D**–**F**) show values for female mice. Data are presented as mean ± SD (*n* = 6 males, 5 females per condition). * *p* < 0.05, ^##^ *p* < 0.01, ** *p* < 0.005.

**Figure 5 biomolecules-14-00426-f005:**
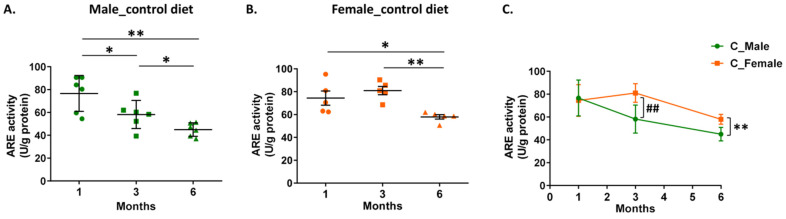
Effect of aging and sex on PON1 arylesterase (ARE) activity in the cortical homogenate of 5xFAD mice fed with the control diet. ARE activity was normalized with total protein concentration and expressed as U/g of protein. Panel (**A**) shows values for male mice, panel (**B**) shows values for female mice, and panel (**C**) shows a comparison of ARE activity between male and female mice. Data are presented as mean ± SD (*n* = 6 males, 5 females per condition). * *p* < 0.05, ^##^ *p* < 0.01, ** *p* < 0.005.

**Figure 6 biomolecules-14-00426-f006:**
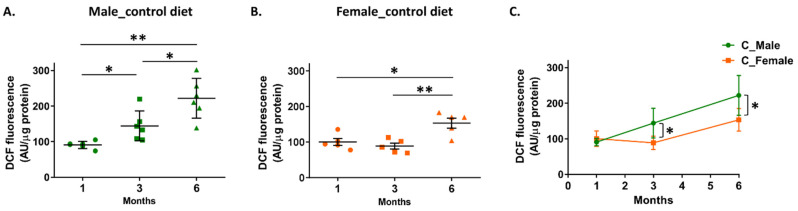
Effect of aging and sex on free radical levels in the cortical homogenate of 5xFAD mice fed with the control diet. Free radical levels, that were monitored by changes in the fluorescence intensity of DCF, normalized with total protein concentration and expressed as arbitrary units (AU)/µg of protein. Panel (**A**) shows values for male mice, panel (**B**) shows values for female mice, and panel (**C**) shows a comparison of DCF fluorescence between male and female mice. Data are presented as mean ± SD (*n* = 6 males, 5 females per condition). * *p* < 0.05, ** *p* < 0.005.

**Figure 7 biomolecules-14-00426-f007:**
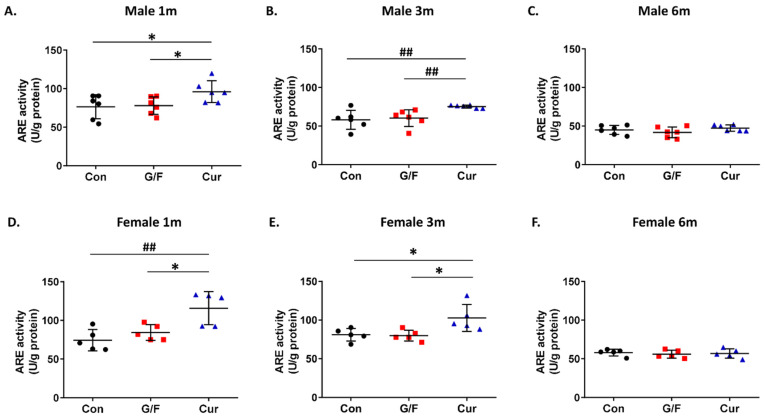
Time-dependent effect of currant diet on PON1 arylesterase (ARE) activity in the cortical homogenate of 5xFAD mice. ARE activity was normalized with total protein concentration and expressed as U/g of protein. Panels (**A**–**C**) show values for male mice and panels (**D**–**F**) show values for female mice. Data are presented as mean ± SD (*n* = 6 males, 5 females per condition). * *p* < 0.05, ^##^ *p* < 0.01.

**Figure 8 biomolecules-14-00426-f008:**
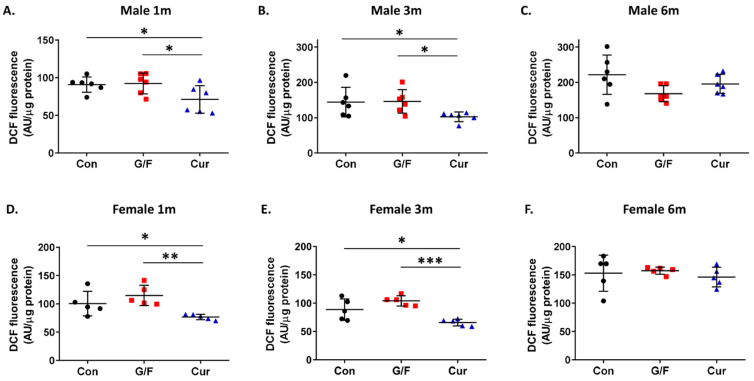
Time-dependent effect of currant diet on free radical levels in the cortical homogenate of 5xFAD mice. Free radical levels, that were monitored by changes in the fluorescence intensity of DCF, normalized with total protein concentration and expressed as arbitrary units (AU)/µg of protein. Panels (**A**–**C**) show values for male mice and panels (**D**–**F**) show values for female mice. Data are presented as mean ± SD (*n* = 6 males, 5 females per condition). * *p* < 0.05, ** *p* < 0.005, *** *p* < 0.0001.

**Figure 9 biomolecules-14-00426-f009:**
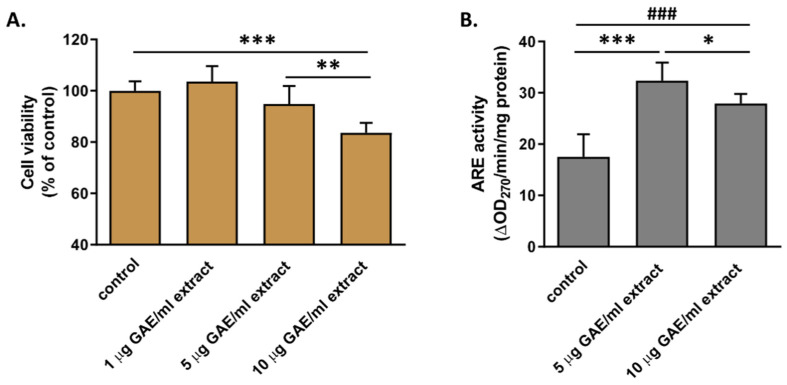
Effect of currant extract on the viability of Huh-7 cells and secreted PON1 activity. (**A**) The survival of Huh-7 cells incubated with increasing concentrations of currant polar phenolic extract, for 48 h, was determined by an MTT assay. Cell viability was expressed as percent relative to the viability of control, untreated cells set to 100%. (**B**) Arylesterase (ARE) activity of PON1 secreted in the medium of cells treated with 5 μg GAE/mL and 10 μg GAE/mL currant polar phenolic extract for 48 h, and then, with medium containing heat-inactivated FBS for 24 more hours. The results were normalized to the protein content of the cells and expressed as ΔOD_270_/min/mg of cell protein. Data are presented as mean ± SD. * *p* < 0.05, ** *p* < 0.005, ^###^ *p* < 0.001, *** *p* < 0.0001.

**Figure 10 biomolecules-14-00426-f010:**
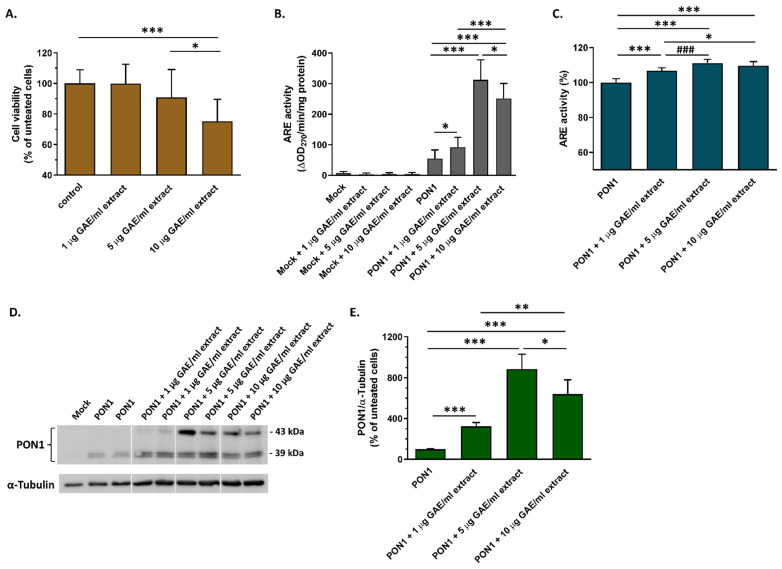
Effect of currant extract on the viability of HEK293 cells, PON1 expression, and secreted PON1 activity following transfection of cells with a PON1-expressing plasmid. (**A**) The survival of HEK293 cells incubated with increasing concentrations of currant polar phenolic extract, for 48 h, was determined by an MTT assay. Cell viability was expressed as percent relative to the viability of control, untreated cells set to 100%. (**B**) Arylesterase (ARE) activity of PON1 secreted in the medium of cells transfected with a pcDNA3.1/C-(K)-DYK/PON1 vector (PON1) or with empty vector (mock) and incubated with increasing concentrations of currant polar phenolic extract for 48 h, and then, with medium containing heat-inactivated FBS for 24 more hours. The results were normalized to the protein content of the cells and expressed as ΔOD_270_/min /mg of cell protein. (**C**) ARE activity of PON1 incubated with increasing concentrations of currant polar phenolic extract in vitro. As an enzyme source we used the cell medium of HEK293 cells transfected with the pcDNA3.1+/C-(K)-DYK/PON1 vector and incubated in the absence of currant extract. ARE activity was expressed as a percent relative to the control activity (measured in the absence of extract) set to 100%. (**D**) Western blot showing the expression of PON1 in the lysate of HEK293 cells transfected with a pcDNA3.1/C-(K)-DYK/PON1 vector (PON1) or with empty vector (mock) and incubated with increasing concentrations of currant polar phenolic extract for 48 h. PON1 was detected using an antibody (9A3) against the DYKDDDDK Tag fused with protein. Cellular α-tubulin was also detected as a loading control. All samples used for the comparative analysis were run on the same gel. White lines indicate splicing together of lanes, following the removal of unnecessary lanes, originating from the same blot image (Appendix A). (**E**) Western blots were scanned and quantified by ImageJ software. The normalized levels of PON1 in cell lysates against α-tubulin are shown as percent relative to the levels of transfected untreated cells set to 100%. Data are presented as mean ± SD. * *p* < 0.05, ** *p* < 0.005, ^###^ *p* < 0.001, *** *p* < 0.0001.

## Data Availability

The data that support the findings of this study are available from the corresponding author (Angeliki Chroni), upon reasonable request.

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
