# Peer review of "Corinthian Currants Promote the Expression of Paraoxonase-1 and Enhance the Antioxidant Status in Serum and Brain of 5xFAD Mouse Model of Alzheimer’s Disease"

_biomolecules, 2024, doi:10.3390/biom14040426_

Round 1

Reviewer 1 Report

Comments and Suggestions for Authors

Dr.  Lymperopoulos et al the effect of nutrition supplementation on PON1 activity in 5xFAD mice. They found the age-dependent decrease in PON1 activity in both male and female 5xFAD mice. Oral supplementation of Corinthian currant (CurD) only increased PON1 activity in serum in 1 month old mice. Interestingly, supplementation of the Corinthian current increased the PON1 activity in cortex tissue in both 1- and 3-month-old mice. In addition, authors found that oxidative stress was increased in 3- and 6-month-old mice compared to 1- month old mice. Supplementation of CurD was able to decrease MDA content in serum only in 1-month old mice. The CurD feeding can decrease ROS generation in cortex tissue in both 1- and 3-month-old mice. They further found that the CurD treatment increased the content of PON1 in cultured cells in a dose-dependent manner. Authors concluded that a CurD treatment can decrease the oxidative stress in 5xFAD mice when the treatment starts in the younger age.

It is a well-written manuscript. The reviewer has some minor comments.

2.14. Statistical Analysis

 There were multiple groups in the current study. Authors also studied the effect of CurD in both male and female mice. Based on experimental design, two-way ANNOVA should be used for statistical analysis. Please explain why un-paired t-test was used in the current manuscript.

Figure 10.

Panel A showed that 10 ug CurD provided optimal protection in cultured cells. However, ARE activity and PON1 protein content were higher in 5 ug CurD treated cells compared to 10 ug CurD treatment. Could authors give an explanation why 5 ug CurD treatment did not significantly decrease the cell death?

There were two bands in PON1 blots. The CurD treatment only increased the 43 KD PON1 isoforms. Was there any functional difference in these two isoforms of PON1?

Reviewer 2 Report

Comments and Suggestions for Authors

The authors of the manuscript entitled "Corinthian currants promote the expression of  paraoxonase-1 and enhance the antioxidant status in serum and the brain of the 5xFAD mouse model of Alzheimer's disease “by the authors Dimitris Lymperopoulos et al., have shown through their findings that a diet supplemented with Corinthian currants in 5xFAD mice of both sexes had increased serum PON1 activity during the early stages of AD. They also showed an inverse correlation of brain PON1 activity with free radical levels with a reduction in free radical levels on the brain of 5xFAD mice suggesting its anti-oxidant role. The authors show the association of PON1 activity in 5xFAD mice older than 2 months and mention that their data is similar to data shown in human studies however, I strongly believe that further studies is necessary at the molecular level. It would be great if the authors could supplement their data with westerns showing the amyloid β levels or tau levels. Furthermore, a detailed analysis with tissue sections of the brain with the new diet of Corinthian currants and a reduction in plaques would be a stronger dataset to show the correlation between PON1 levels and AD. It would be great if the authors could supplement their manuscript with such data. Also, with respect to the original blots provided by the authors the image PON1 06.10.23_26 inv.jpg does not match with the image represented in the main figures. Overall, I strongly believe that the authors can increase the significance of their study by complementing with important brain related datasets.

Round 2

Reviewer 1 Report

Comments and Suggestions for Authors

My concerns have been properly addressed.

Author Response

Thank you for the positive evaluation of our manuscript.

Reviewer 2 Report

Comments and Suggestions for Authors

The author's of the manuscript have provided sufficient evidence with regards to my questions and concerns raised.

However, with regards to my second comment the authors should draw an outline around the cropped regions of the blot. The current representation means that the samples were run on a  whole/complete blot non manipulated blot without any cropping of images.

Overall, the manuscript is good to be published.

Author Response

We have now added a white dividing line to indicate the cropping of blots in Figure 10D. An explanation has been added in the legend of Figure 10 as follows: “All samples used for the comparative analysis were run on the same gel. White lines indicate splicing together of lanes, following the removal of unnecessary lanes, originating from the same blot image.”